# Comparative Study of the Use of Doxycycline and Oxytetracycline to Treat Anaplasmosis in Fattening Lambs

**DOI:** 10.3390/ani12172279

**Published:** 2022-09-02

**Authors:** Delia Lacasta, Héctor Ruiz, Aurora Ortín, Sergio Villanueva-Saz, Agustín Estrada-Peña, José María González, Juan José Ramos, Luis Miguel Ferrer, Alfredo Ángel Benito, Raquel Labanda, Carlos Malo, María Teresa Verde, Antonio Fernández, Marta Ruiz de Arcaute

**Affiliations:** 1Animal Pathology Department, Instituto Agroalimentario de Aragón-IA2 (Universidad de Zaragoza-CITA), Veterinary Faculty of Zaragoza, C/Miguel Servet 177, 50013 Zaragoza, Spain; 2Gabinete Técnico Veterinario S.L., C/Isla Conejera sn, 50013 Zaragoza, Spain; 3Exopol S.L. Pol. Río Gállego D-8, San Mateo de Gállego, 50840 Zaragoza, Spain; 4Casa de Ganaderos de Zaragoza, C/San Andrés, 8, 50001 Zaragoza, Spain; 5A.D.S. Ayerbe la Sotonera, 22005 Huesca, Spain

**Keywords:** anaplasmosis, lambs, *Anaplasma ovis*, jaundice, carcass condemnation, oxytetracycline, doxycycline

## Abstract

**Simple Summary:**

Ovine anaplasmosis has been causing relevant economic losses during the last years due to icteric lamb carcasses condemnation. They are apparently healthy lambs in good body condition that, after being slaughtered, show an icteric coloration of their carcasses. Anaplasmosis produces strong haemolysis that leads to an increase in bilirubin that stains the carcasses. The present work analyzes the therapeutic effect of oxytetracycline and doxycycline in *Anaplasma ovis* infected fattening lambs.

**Abstract:**

Lamb icteric carcasses condemnation due to *Anaplasma ovis* is causing relevant economic losses. A comparative study was developed on the effects of different antibiotics to treat ovine anaplasmosis in fattening lambs. A total of 100 *A. ovis* naturally infected lambs were selected and randomly divided into four groups of 25 lambs: Group ID, treated with injectable doxycycline; Group OD, oral doxycycline; Group O, injectable oxytetracycline; and Group C, untreated animals for the control group. Clinical, haematological, and molecular analyses were performed before the treatment and 12 and 45 days after the beginning of the treatments, and carcass condemnation was followed after slaughter. The *A. ovis* bacterial load was high before the treatments in the four groups and decreased significantly 45 days after treatment in the ID and O Groups (*p* < 0.001). The parameters that were related to haemolysis showed similar results. At the abattoir, 15 out of the 47 examined carcasses were condemned; 7 of C Group, 6 of OD Group, 2 of O Group, and 0 of ID Group. It can be concluded that injectable doxycycline and oxytetracycline significantly reduce *A. ovis* bacterial load in blood and carcass condemnation at the abattoir. Further studies are needed in order to confirm these encouraging findings.

## 1. Introduction

Ovine anaplasmosis, an emerging disease in Europe, is being spread quickly through Spanish sheep flocks, causing relevant economic losses. This disease was described in tropical and subtropical countries as a mild disease that did not produce relevant clinical signs. However, when this bacterium enters a naïve population, the clinical picture can be much more severe [1]. In Spain, *A. ovis* infection was reported for the first time in wild ruminants, in roe deer (*Capreolus capreolus*) [2]. Since 2014, when an outbreak of ovine anaplasmosis was diagnosed for the first time in Spain [3], the number of severe clinical cases of anaplasmosis has increased considerably. 

*Anaplasma ovis* is an obligate intraerythrocytic Gram-negative bacteria that belongs to the family Anaplasmataceae. This species is primarily transmitted by ticks, particularly the genera *Ixodes*, *Dermacentor*, *Rhipicephalus*, and *Amblyomma* and can infect sheep, goats, and wild ruminants [4,5,6]. *Anaplasma* genus bacteria are transmitted by mechanical and biological vectors, mainly ticks. Some studies have shown that there are up to 19 types of ticks that are capable of transmitting the disease [4], depending on geographical location and seasonality. In Spain, the presence of *Anaplasma* bacteria in ticks of the *Rhipicephalus*, *Ixodes*, *Hyaloma*, *Dermacentor*, and *Amblyomma* genera has been demonstrated [6]. However, ticks of the genus *Dermacentor (D. silvarum*, *D. marginatus*, *D. andersoni)* and *Riphicephalus (Rhipicephalus bursa)* are the main biological vectors of anaplasmosis [5]. In the salivary glands of ticks, bacteria replicate, increasing infective capacity [7]. In other *Anaplasma* species, infectivity is achieved when there are 10^6^ microorganisms in ticks’ salivary glands [8]. It is because of this dose-dependence that biological transmission has greater efficacy and, therefore, greater relevance compared to the mechanic. The bacterium replicates by binary fission within the erythrocyte and leaves it using a not well-defined mechanism, but apparently not lytic, to infect new erythrocytes [9]. Thus, the haemolytic anaemia that is associated with this disease results from the immune response. The activated macrophages destroy infected erythrocytes through opsonization and nitric oxide production [10]. During the acute phase, the number of infected erythrocytes doubles every 24–48 h [11]. However, this is a chronic process, and the destruction of the erythrocytes is slow and progressive, and the animal begins to show noticeable changes around 30–40 days after infection [3]. 

Depending on the naïve immune system, age, and other still unknown factors, there are animals where anaemia goes unnoticed and others in which this disease causes severe clinical signs. The acute phase of the disease is characterized by nonspecific weakness, depression, a marked loss of body condition, fever peaks, progressive anaemia, and finally, the death of the animals due to secondary infections [3,12,13]. Severely affected animals that do not die often have to be culled due to lack of production. There are flocks where the number of affected animals can be very high, causing significant economic losses, especially since the most commonly affected sheep are yearlings [3,13]. The infected animals that survive the infection remain infected for life, maintaining a high bacterial load in the blood for at least seven years (unpublished data).

Until 2020, the disease had only been diagnosed in adult sheep in Spain, with the highest incidence in one or two-year-old animals. However, in 2020, an outbreak of ovine anaplasmosis was diagnosed for the first time in fattening lambs [13]. After the slaughter of apparently healthy three-month-old lambs, 34.84% showed jaundiced carcasses. All of the tested lambs with icteric carcasses had severe regenerative anaemia just before slaughter and showed positive *A. ovis* PCR with a high bacterial load in the blood. During the spring of 2020, similar clinical cases were diagnosed in different geographical areas of Spain, reaching global percentages of close to 2% of lamb carcasses condemnation due to jaundice at the Mercazaragoza slaughterhouse (Zaragoza, Spain), causing very relevant economic losses (personal communication). In the spring of 2021, the cases of jaundiced carcasses condemnations were again diagnosed at the slaughterhouse, reconfirming the presence of *A. ovis* in affected animals. The percentage of condemnation in affected flocks is usually close to 40% during late spring.

Based on the substantial economic losses that this disease is causing the affected farmers, it was decided to conduct a study about possible antibiotic treatments to control anaplasmosis in affected lambs. The obtained data are presented in the current paper.

## 2. Materials and Methods 

In April 2021, a clinical case of icteric carcasses condemnation affecting fattening lambs was referred to the Ruminant Clinical Service of the Veterinary Faculty of Zaragoza. The sheep farm was affected by anaplasmosis in the previous years. 

The affected farm was a 1000 sheep meat farm that was managed under an extensive production system and located in Apies, north of the Aragon region, at 680 meters above sea level with an annual average rainfall of 670 mm per year (Figure 1). 

The farmer raised Rasa Aragonesa breed sheep and produced lambs with a protected geographical indication (PGI) “Ternasco de Aragón” (two- to three-month-old lambs with 21 to 23 kg of live weight at slaughter). Its reproductive management consisted of four 45-days-long matting periods per year. Although the dams grazed outdoors, the lambs were kept indoors until slaughter. The ewes went out to graze in the morning and returned to the farm in the late afternoon. The lambs were maintained on the farm with straw, compound feed, and water ad libitum until the arrival of their dams. The lambs were weaned when they were 45 days old and kept on the farm until slaughter.

Attending the clinical case, clinical, haematological and molecular studies were carried out on 156 lambs. Severe regenerative anaemia with monocytosis was observed in 48% and 100% of the animals, respectively. Molecular analyses showed positive results for *A. ovis* in 116 of the studied lambs (116/156: 74.35%), confirming that we were again facing a case of anaplasmosis in fattening lambs [11].

Then, it was decided to conduct a study on 100 lambs with different antibiotic treatments to curb the disease’s incidence in these lambs and observe which antibiotic treatment offered the best results.

All of the procedures were conducted under Project Licence PI 43/19 and approved by the Ethics Committee for Animal Experiments from the University of Zaragoza. The care and use of animals were performed according to the Spanish Policy for Animal Protection RD53/2013, meeting the European Union Directive 2010/63 on the protection of animals that are used for experimental and other scientific purposes.

### 2.1. Studied Lambs

A total of 100 recently weaned (40 days old) *A. ovis*-positive and apparently healthy lambs were selected for the study. The lambs were randomly divided into four groups of 25 lambs each and weighed (15 kg on average) to calculate the dose of antibiotic to be administered. The twenty-five lambs of Group ID were treated with injectable doxycycline every 24 h for 7 days intramuscularly (DFV Doxivet injectable. DIVASA FARMAVIC (Barcelona, Spain), S.A.; 10 mg of doxycycline/kg BW/day). Lambs belonging to Group OD were treated with oral doxycycline every 24 h for 10 days (DVF DOXIVET 100/10 mg/mL. DIVASA FARMAVIC, S.A. (Barcelona, Spain) 10 mg doxycycline (hyclate)/kg BW/day + 1 mg bromhexine/Kg BW/day). The dose was administered individually to each lamb every day. The 25 lambs of the third group (Group O) were treated intramuscularly three times with oxytetracycline every 56 h (Forticlina retard 200 mg/mL. SYVA, S.A.U. (León, Spain) 20 mg of oxytetracycline/kg BW). Finally, 25 lambs were kept as a control group, and no antibiotic treatment was applied (Group C). During the analyzed period, one of the lambs of the O group died due to an ovine respiratory complex and was removed from the study.

### 2.2. Clinical Examination and External Parasites Assessment 

Clinical examination was performed on the selected lambs before applying the antibiotic treatments. During the examination, several clinical parameters were recorded: body condition, the color of mucous membranes and respiratory, and digestive or locomotor clinical signs.

The farmers reported that the number of ticks had increased in the last years during the spring. They referred that even lambs, reared indoors, were massively parasitized with ticks. During the clinical examination, the presence of ticks in the lambs was checked, and this assertion was confirmed. An appropriate sample of more than 50 ticks was collected from the lambs to study the genus and species to which the ticks belonged.

### 2.3. Biomolecular Analysis

Whole blood samples from the 100 studied lambs were submitted to the laboratory to detect *A. ovis* by qPCR in three moments of the study: T0, previously to the treatment; T1, 12 days; and T2, 45 days after the beginning of the treatment. *A. ovis* was detected using the commercial kit EXOone *Anaplasma ovis* (EXOPOL S.L., San Mateo de Gállego, Spain) and following the manufacturer’s instructions. This qPCR assay has an analytical sensitivity of 50 copies of genomic equivalent/reaction and includes a quantified synthetic positive control. The assay targets the single copy MSP4 gene that is reported to allow a specific differentiation of *A. ovis* from the highly related *Anaplasma marginale* [14]. An endogenous control was also included in these assays to avoid false-negative results. The bacterial load was expressed using the quantification cycle (Cq), which is the cycle number where the PCR amplification curve intersects the threshold line [15]. The Cq value can be used to quantify or determine the presence/absence of the target sequence.

The commercial kit, MagMAX™ Pathogen RNA/DNA (Thermo Fisher Scientific, Austin, TX, USA) with an automated magnetic particle processor (KingFisher Flex System, Thermo Fisher Scientific, Vantaa, Finland), was used for nucleic acids extraction according to the manufacturer’s instructions. Amplification was carried out in a QuantStudio 5 Real-Time PCR machine (Applied Biosystems, Marsiling, Singapore), and results were analyzed with the respective software (QuantStudio Design & Analysis software v1.5.1).

### 2.4. Haematological Analysis

Blood samples with anticoagulant (EDTA) were collected from the jugular vein through a vacutainer system from the studied animals (n = 100) in T0, previously to the treatment; T1, 12 days; and T2, 45 days after the beginning of the treatment. As one lamb died after the first sampling, it was removed from the experiment, with 99 being the final studied number of lambs. Haematology was performed using an automatic haematological counter IDEXX Procyte Dx (IDEXX laboratories, Westbrook, ME, USA). The measured parameters were leucocytes (K/µL), erythrocytes (M/µL), haemoglobin (g/dL), haematocrit (%), platelets (K/µL), MCV (mean corpuscular volume; fL), MCH (mean corpuscular haemoglobin; pg), MCHC (mean corpuscular haemoglobin concentration; g/dL), and reticulocytes (K/µL). Moreover, neutrophils (K/µL), lymphocytes (K/µL), monocytes (K/µL), basophils (K/µL), and eosinophils (K/µL) were analyzed in the white blood cells. 

### 2.5. Carcasses Examination

The lambs were weighed individually every week, and when they reached slaughter weight (21 to 23 kg.), they were sent to the slaughterhouse. A follow-up of 47 carcasses and viscera could be carried out during the slaughtering out of 100 studied lambs, taking data on the icteric color of the carcasses and condemnations. A total of 12 belonged to the control group, 14 to the OD group, 17 to the O group, and 4 to the ID group.

### 2.6. Statistical Analysis

The treatment groups were used as grouping variables for all the results. Descriptive statistics based on counts and proportions were used to define qualitative variables: clinical parameters, carcass jaundice, and condemnations. The Chi-square test was used for haematological parameters, with Fisher correction when some cells showed less than 5 cases. The median value and 25th and 75th percentile values were used to show the result of the quantitative variable (qPCR and haematological parameters). Non-parametric tests were performed for analyzing these variables because they did not meet the normality criteria. The procedure followed was the Friedman Test for related samples in each treatment. When this test was statistically significant, the differences were analyzed using the Wilcoxon signed-rank test for two related samples. In addition, the differences between treatments were analyzed for each sampling time according to the Kruskal–Wallis test. When this test was statistically significant, the differences were analyzed using the Mann–Whitney U test for two independent samples. For all cases, *p* < 0.05 was required to consider statistically significant differences. These tests were developed with IBM SPSS Statistics V.24 software.

## 3. Results 

### 3.1. Clinical Examination and External Parasites Assessment 

All the lambs in the study showed a healthy appearance, a good body condition score, and an expected growth for this breed throughout the experiment. No clinical signs were observed, and only one lamb died due to a respiratory disorder during the study period. The others reached the abattoir with the average weight. No lamb showed jaundice at any time.

During the examination of the lambs, a massive tick parasitization was observed. Most of the ticks were found to adhere to the inner face of the pinna. All of the ticks that were collected from the lambs belonged to the genus *Riphicephalus sanguineus* sensu lato. 

### 3.2. Biomolecular Analysis

Table 1 shows the qPCR results expressed as Cq median values of the lambs belonging to the four treatment groups during the three moments of the study in which they were analyzed: before treatment (T0), 12 days (T1), and 45 days post-treatment (T2).

As can be observed in Table 1, the *A. ovis* bacterial load before the treatment (T0) was high (low Cq value) in the four groups, and no statistical differences were observed between them. However, the bacterial load decreases significantly 12 days after treatment (T1) in all of the antibiotic-treated groups, remaining stable in the control group (*p* < 0.001). Finally, 45 days after treatment, the bacterial load increased T2 vs. T1 in the OD group (*p* < 0.001), reaching similar levels to the control group. The groups that were treated with injectable doxycycline (ID) and oxytetracycline (O) continued to reduce the bacterial load 45 days after treatment (significantly in the case of the ID group; *p* < 0.001). Nevertheless, all the animals remained positive for *A. ovis* throughout the experiment.

### 3.3. Haematological Analysis

Those blood parameters that were related to the appearance of jaundice that were associated with hemolytic anaemia (erythrocytes, haematocrit and reticulocytes) were deeply analyzed, as well as monocytes, which were altered in previous studies that were carried out on lambs that were affected by anaplasmosis [11]. The other haematological parameters were analyzed but did not show significant alterations.

#### 3.3.1. Erythrocytes

The number of erythrocytes did not present significant differences between the groups before the beginning of the treatments (Table 2). Some animals showed mild anaemia with erythrocyte levels that were slightly below the normal values (9.49–15.12 M/μL). However, 12 days after treatment (T1), the animals in the control group decreased the number of erythrocytes, while all the treated groups increased the erythrocytes recounts (*p* < 0.001 for the three groups). Significant differences were observed between the control and the three treated groups at T1 (*p* < 0.001 for the three groups). In addition, the lambs that were treated with injectable doxycycline (ID) showed a higher erythrocyte rate 12 days after treatment than the OD group (*p* < 0.027). At 45 days post-treatment, the control group continued with low recounts, showing significant differences in the erythrocyte count with the treated animals (O, OD, and ID, *p* < 0.001 in all cases). Similarly to T1, the animals that were treated with injectable doxycycline (ID) maintained significant differences with the OD group (*p* < 0.001).

Interestingly. the animals in the control group showed an apparent worsening over time. although no significant differences were found (*p* > 0.001). Those in the O and ID groups improved throughout the analyzed period. showing significant differences between the sampling times. However. lambs belonging to the OD group improved in the first sampling (*p* = 0.003). but they remained stable in the second sampling with respect to the previous one (*p* = 0.835).

#### 3.3.2. Haematocrit

Although all the lambs remained within the reference values (27.0–42.0%) throughout the study. differences between the groups with the same trend as erythrocytes were still seen in the haematocrit value (Table 3). A worsening of the control group and an apparent improvement in the treated lambs were observed.

#### 3.3.3. Reticulocytes

The differences in reticulocytes parameter showed that the control group maintained a high level of reticulocytes throughout the study (threshold values 0.0–15.0 K/µL). However, in the lambs of the three treated groups. the reticulocyte count drastically decreased at T1 (*p*< 0.001) (Figure 2). Both the ID and O groups continued to reduce the numbers of reticulocytes 45 days after treatment compared to the C group (*p* = 0.003). However, the OD group in T2 showed higher values than the O group (*p* = 0.025) and similar to the ID group (*p* = 0.056) and C group (*p* = 0.269).

#### 3.3.4. Monocytes

One of the lambs of the O group and two of the OD group lack the count of this parameter due to technical issues with the haematological counter in the last sampling and. for that reason. were removed from monocytes statistical analysis. 

Table 4 displays all the lambs that showed a clear monocytosis (threshold values 0.1–1.01 K/μL) at the beginning of the study. which was corrected at 12 days in the OD- and ID-treated groups (*p* < 0.001). increasing again in the last sampling. The control group maintained high values of monocytes throughout the study. increasing significantly in the last sampling (*p* < 0.001). 

At T1. the three treated groups presented lower monocyte values than the C group (*p* = 0.002. *p* < 0.001, and *p* < 0.001 for O. OD, and ID. respectively). Something similar happened in the T2 monocyte values; O. OD, and ID showed lower values than the C group (*p* < 0.001. *p* = 0.004, and *p* < 0.001 for O. OD, and ID. respectively). There were no differences between the treated groups at any sampling.

### 3.4. Carcasses Examination

At the abattoir. 15 out of the 47 examined carcasses were condemned due to jaundice. A total of seven belonged to the control group (7/12:58.33%). six to the oral doxycycline group (6/14: 42.85%), and two to the oxytetracycline group (2/17: 11.76%). No carcass of the four lambs that were treated with injectable doxycycline that could be followed at the abattoir were condemned or showed jaundice.

## 4. Discussion

Ovine anaplasmosis is an emerging disease in Europe [1,16,17] that is becoming of relevant economic importance in the sheep industry. Since it was diagnosed for the first time in Spain [13]. icteric lamb carcasses condemnations due to *A. ovis* have become the leading cause of carcass condemnation in some Spanish abattoirs (personal communication).

Ovine anaplasmosis is a disease that can cause very diverse clinical pictures depending on the animal’s immune system, its body condition, environmental conditions, etc. [1,12,18]. Anaemia that is caused by anaplasma can become severe. but its clinical manifestations may be variable [3]. In recent years. it has been observed that if the lambs become infected during the lactation period, they will be just at the peak of haemolysis, which is produced 30 days after infection [3,13], when they are slaughtered a month or a month and a half later. This being responsible for the appearance of icteric carcasses and consequent condemnation. 

Oxytetracycline is the only antibiotic that is licensed for anaplasmosis in cattle in Spain, and tetracyclines have been demonstrated to be effective for treating active anaplasmosis. However. their ability to eliminate Anaplasma at currently approved therapeutic doses remains unclear [19]. For that reason, it was decided to apply this antibiotic to one of the analyzed groups. The other two groups were treated with doxycycline, the antibiotic of choice in dogs and humans [20,21,22] and the best option based on our previous personal experience. Doxycycline was administered intramuscularly in one group and orally in the other. The group of oral doxycycline was included to offer easier handling for farmers if good results were obtained. The treated lambs were preruminants; consequently, there was no risk of rumen indigestion from oral antibiotics.

It is well known that the main biological vectors of anaplasmosis are ticks belonging to the genera *Rhipicephalus* (i.e. *Rhipicephalus bursa* and *Rhipicephalus sanguineus* sensu lato); *Dermacentor* (i.e. *Dermacentor silvarum. Dermacentor marginatus.* and *Dermacentor andersoni); Ixodes;* and *Amblyomma* [4,5,6,23,24], according to the different species of Anaplasma and the biogeographic region. In the present survey, as in the study that was carried out by our team in 2020 [13], the only tick species that was found was *R. sanguineus* s.l. [25]. It was observed that although the lambs were raised intensively without grazing. their dams brought the ticks into the shed after going out to graze and were responsible for the transmission of *A. ovis* to their lambs. According to this, it was concluded that the relationship between seeing an increase in the presence of ticks in the flock and the presentation of severe anaplasmosis showed a significant correlation [18].

The *A. ovis* bacterial load in blood has been shown to be decisive in determining the degree of anaemia of the animal as well as the severity of the clinical signs [3,13]. During the acute phase of the disease, the number of infected erythrocytes doubles every 24–48 h [11]. In the study that was performed by Jimenez et al., the *A. ovis* experimentally-infected animals showed that the severe anaemia that was observed after the first delivery coincided with a peak in the bacterial load and a relevant decrease in the humoral response [3]. Likewise, the *A. ovis* Cq mean of samples that were obtained from the infected animals while suffering anaemia was significantly lower than Cq from samples when the animals did not show anaemia [3]. This was in accordance with the study that was performed by our group on fattening lambs in 2020 [13]. In that survey. all of the tested lambs with icteric carcasses showed positive *A. ovis* PCR, and the bacterial load was significantly higher in the animals that showed jaundiced carcasses. In the current survey, *A. ovis* bacterial load was high before the treatment in all the studied lambs, and no statistical differences were observed between the groups. However, the bacterial load decreased significantly 12 days after treatment in all antibiotic-treated groups, and those animals that were treated with injectable doxycycline and oxytetracycline remained in low Cq values throughout the study. although they never showed negative results to *A. ovis*. Nonetheless, the lambs that were treated with oral doxycycline increased again in the last sampling, reaching similar levels to the control group. 

The erythrocyte count, the haematocrit value, and the reticulocytes are the main parameters that are used to measure the degree and type of anaemia. During the acute phase of ovine anaplasmosis. affected animals usually present severe regenerative anaemia, and these values are highly altered with a drastic reduction in the red cell count and an increase in reticulocytes [1,12,13]. In the present work, the levels of these parameters before applying the treatments showed mild regenerative anaemia in 49% of the animals. All of the lambs that were treated with antibiotics recovered at normal rates 12 days after treatment. However, although lambs that were treated with injectable doxycycline and oxytetracycline continued to improve, those that were treated with oral doxycycline remained stable or worsened 45 days after the beginning of the treatment.

After cell infection, anaplasma leaves the erythrocyte using a not well-defined mechanism, apparently non-lytic, to infect new erythrocytes. Then, the severe haemolytic anaemia that is associated with this disease is a consequence of the immune response that causes anaplasma in the body. After antigen presentation, CD4 + Th1-lymphocytes produce IFN-y, inducing the production of IgG2, which, in coordination with activated macrophages, are capable of destroying infected erythrocytes through opsonization and nitric oxide production [10]. In addition, in previous studies, it has been observed that monocytosis is a common finding in acute anaplasmosis [13]. In the current survey, all the lambs showed monocytosis before treatment, which was corrected 12 days post-treatment in the ID- and OD-treated groups.

Finally, at the abattoir, 58.33% of the carcasses of the control group were condemned due to jaundice. In previously published cases of ovine anaplasmosis in fattening lambs, the final percentage of icteric carcasses condemnation reached 34.80% [13], relatively lower than those occurring in the untreated group. Surprisingly, lambs that were treated with oral doxycycline also had a high percentage of condemnations at the slaughterhouse (42.85%). According to previous studies [3,13], this is in agreement with the high bacterial load that these lambs presented in the last sampling, 45 days after treatment. However, only two of the lambs that were treated with oxytetracycline were condemned (11.76%), and none of those that were treated with injectable doxycycline from those that were examined at the abattoir.

Tetracyclines are one of the antimicrobial groups that are authorized for use in veterinary medicine, and they are widely used due to their broad spectrum of activity and limited adverse effects. Doxycycline is a tetracycline that shows high oral bioavailability and distributes intracellularly even to sites that are protected by diffusion barriers [26]. Interestingly, apparent differences were observed depending on the administration route of doxycycline in our survey. The oral administration of antibiotics in ruminants is always controversial, and the lower bioavailability of antibiotics, when administered orally, has been demonstrated [27]. However, the lambs in our study were 1.5 months old and can be considered preruminants. Furthermore, previous investigations revealed differences in the pharmacokinetics of doxycycline between young calves and mature ruminants [28]. In a comparative study that was carried out on the pharmacokinetics of orally administered doxycycline in lambs and lactating ewes, it was observed that the values of clearance were significantly higher in adult sheep and that the plasma concentration was higher in lambs [29]. However, in calves, it was observed that after intravenous doxycycline administration. the values of clearance in animals with immature rumen function were higher than in calves with mature function [28]. Further, previous studies have also shown that age and diet can affect the disposition of some antibiotics with. for example, a slight decrease in sulphonamides bioavailability that is observed in animals that are fed with grain as compared to animals that are receiving milk [30]. Further studies and new investigations are desirable to understand the differences that were found in the results of doxycycline that was administered intramuscularly or orally to treat anaplasmosis in our survey.

Although doxycycline is the antibiotic of choice for anaplasmosis in other species and the good results observed in our study if it is used intramuscularly, doxycycline is not licensed for ovine in any European country. Then, a cascade prescription is needed to be used after the registered antibiotics have been applied and proven ineffective. As reducing antibiotic use in livestock has become a priority for the management of antimicrobial resistance risk, there is a need for new records of antibiotics in sheep, such as doxycycline to treat anaplasmosis.

## 5. Conclusions

Finally, based on the results that were obtained in the current experiment, it can be concluded that both oxytetracycline and doxycycline have a clear effect in reducing the bacterial load of *A. ovis* in blood in fattening lambs, and as a consequence, in the associated clinical signs. However, when doxycycline is administered orally, it does not seem to reach sufficient therapeutic levels to control the replication of the bacteria, with a worsening of the animals observed 45 days after treatment and a high percentage of condemnations at the abattoir. Thus, we can conclude that injectable oxytetracycline and doxycycline treatments effectively controlled *A. ovis* infection for at least 45 days. They reduced jaundice in lambs and decreased the economic losses that were associated with ovine anaplasmosis in fattening lambs. Nevertheless, further studies are needed in order to confirm these encouraging findings.

## Figures and Tables

**Figure 1 animals-12-02279-f001:**
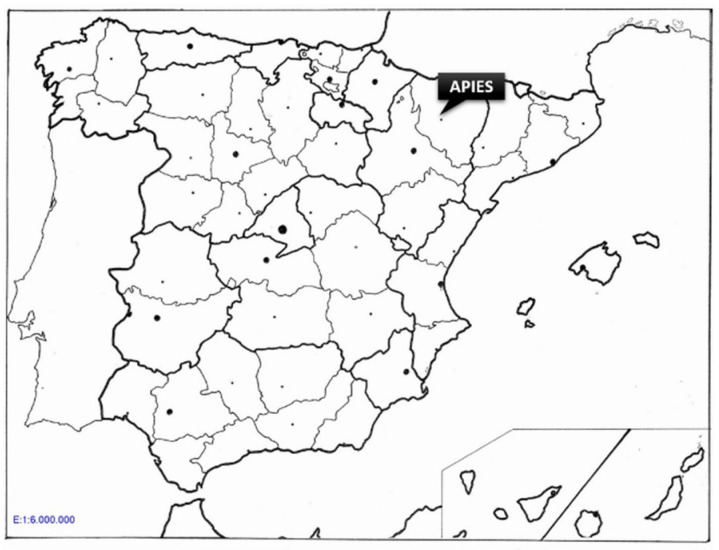
Location of the studied farm in Apies, Aragón region, north of Spain.

**Figure 2 animals-12-02279-f002:**
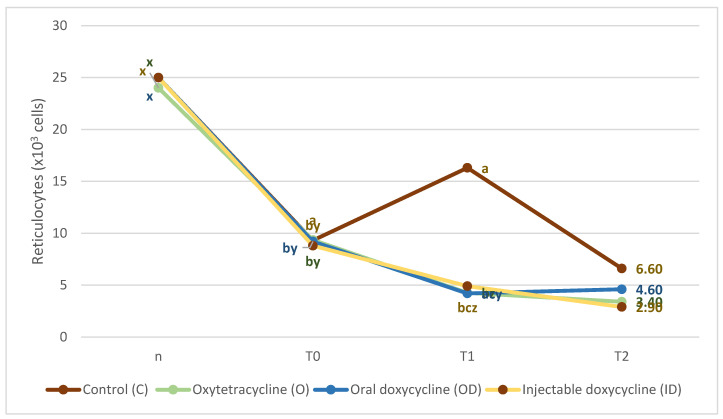
Median (25th and 75th percentile values) reticulocytes (K/µL) of lambs that were treated with oral doxycycline (OD). injectable doxycycline (ID). oxytetracycline (O), and untreated animals belonging to the control group (C) in the three moments that were analyzed: before treatment (T0). 12 days after treatment (T1), and 45 days after treatment (T2). Reticulocyte threshold values are between 0.0 and 15.0 K/µL. ^abc^ indicate significant differences (*p* < 0.005) between treatments at the same sampling time, and ^xyz^ indicate significant differences (*p* < 0.005) between the different sampling times within the same treatment.

**Table 1 animals-12-02279-t001:** qPCR results (Cq median value and 25th and 75th percentile values) of the lambs that were treated with oral doxycycline (OD), injectable doxycycline (ID), oxytetracycline (O), and untreated animals belonging to the control group (C) in the three moments that were analyzed: before treatment (T0), 12 days after the beginning of the treatment (T1), and 45 days after the beginning of the treatment (T2).

Group	n	T0	T1	T2
Control (C)	25	25.00 ^x^ (24.000–27.000)	24.00 ^ay^ (23.000–25.500)	25.00 ^ax^ (23.000–25.000)
Oxytetracycline (O)	24	29.50 ^x^ (23.500–33.750)	31.00 ^by^ (31.000–35.000)	34.00 ^by^ (25.300–36.250)
Oral doxycycline (OD)	25	26.00 ^x^ (23.000–30.500)	31.00 ^by^ (30.000–33.000)	24.00 ^ax^ (24.000–25.000)
Injectable doxycycline (ID)	25	24.00 ^x^ (22.000–27.000)	31.00 ^by^ (30.000–32.000)	34.50 ^bz^ (33.000–36.250)

^ab^ indicate significant differences (*p* < 0.005) between treatments at the same sampling time, and ^xyz^ indicate significant differences (*p* < 0.005) between the different sampling times within the same treatment.

**Table 2 animals-12-02279-t002:** Median (25th and 75th percentile values) erythrocyte levels (M/μL) of lambs that were treated with oral doxycycline (OD), injectable doxycycline (ID), oxytetracycline (O), and untreated animals belonging to the control group (C) in the three moments that were analyzed: before treatment (T0), 12 days after treatment (T1), and 45 days after treatment (T2). The erythrocytes threshold values are between 9.49 and 15.12 M/μL.

Group	n	T0	T1	T2
Control (C)	25	8.25 (6.610–11.515)	8.01 ^a^ (6.505–9.625)	7.14 ^a^ (6.150–8.330)
Oxytetracycline (O)	24	9.95 ^x^ (8.193–11.050)	10.52 ^bcy^ (9.625–11.653)	11.99 ^bcz^ (10.760–12.530)
Oral doxycycline (OD)	25	8.90 ^x^ (7.275–10.400)	10.17 ^by^ (8.945–11.365)	10.45 ^by^ (7.980–11.505)
Injectable doxycycline (ID)	25	10.53 ^x^ (6.725–11.680)	11.61 ^cy^ (9.675–12.735)	11.90 ^cz^ (11.055–12.675)

^abc^ indicate significant differences (*p* < 0.005) between the treatments at the same sampling time, and ^xyz^ indicate significant differences (*p* < 0.005) between the different sampling times within the same treatment.

**Table 3 animals-12-02279-t003:** Median (25th and 75th percentile values) haematocrit (%) of lambs that were treated with oral doxycycline (OD). injectable doxycycline (ID). oxytetracycline (O), and untreated animals belonging to the control group (C) in the three moments that were analyzed: before treatment (T0). 12 days after treatment (T1), and 45 days after treatment (T2). Haematocrit threshold values are between 27.0 and 42.0%.

Group	n	T0	T1	T2
Control (C)	25	28.60 ^ax^ (22.600–31.850)	31.00 ^ay^ (27.650–31.950)	27.50 ^axy^ (24.700–30.600)
Oxytetracycline (O)	24	25.60 ^bx^ (18.775–28.600)	33.00 ^by^ (31.225–34.550)	35.50 ^bcz^ (29.600–39.200)
Oral doxycycline (OD)	25	27.10 ^abx^ (23.450–30.150)	37.30 ^cy^ (33.900–38.850)	34.20 ^bz^ (28.700–37.150)
Injectable doxycycline (ID)	25	28.40 ^ax^ (26.050–33.400)	38.20 ^cy^ (34.900–42.700)	38.80 ^cy^ (33.700–41.900)

^abc^ indicate significant differences (*p* < 0.005) between the treatments at the same sampling time, and ^xyz^ indicate significant differences (*p* < 0.005) between the different sampling times within the same treatment.

**Table 4 animals-12-02279-t004:** Median (25th and 75th percentile values) monocytes (K/µL) of lambs that were treated with oral doxycycline (OD). injectable doxycycline (ID). oxytetracycline (O), and untreated animals belonging to the control group (C) in the three moments that were analyzed: before treatment (T0). 12 days after treatment (T1), and 45 days after treatment (T2). The monocyte threshold values are between 0.1 and 1.01 K/μL.

Group	N	T0	T1	T2
Control (C)	25	2.00 ^x^ (1.280–2.960)	1.99 ^ax^ (1.145–2.725)	2.77 ^ay^ (2.200–3.490)
Oxytetracycline (O)	23	1.84 (1.030–2.350)	1.14 ^b^ (0.788–1.490)	1.38 ^b^ (1.230–2.240)
Oral doxycycline (OD)	23	1.86 ^x^ (1.340–3.060)	0.95 ^by^ (0.505–1.280)	1.91 ^bx^ (1.400–2.440)
Injectable doxycycline (ID)	25	1.66 ^x^ (1.270–2.430)	0.93 ^by^ (0.800–1.310)	1.61 ^bx^ (1.290–2.075)

^ab^ indicate significant differences (*p* < 0.005) between the treatments at the same sampling time, and ^xy^ indicate significant differences (*p* < 0.005) between the different sampling times within the same treatment.

## Data Availability

The data that support the findings of this study are available from the corresponding author upon request.

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
