# Peer review of "Comparative Study of the Use of Doxycycline and Oxytetracycline to Treat Anaplasmosis in Fattening Lambs"

_animals, 2022, doi:10.3390/ani12172279_

Round 1
Reviewer 1 Report
Overall, a well-executed study that has practical and technical value for those working in animal health and agriculture in regions affected by ovine anaplasmosis. Well done. I do have a few suggestions for additions that will improve the manuscript.
General comments
1. Please include the reference ranges for hematocrit, reticulocytes, and monocytes in the text of the manuscript, not just the figure legends.
2. Please be specific in the route of administration of "injectable" drugs. In your discussion you finally specify that the route is IM, but, in much of the manuscript, the route is simply listed as "injectable" which is too general for your readers.
3. Throughout much of the manuscript, you have been careful to not write a percentage for data obtained at slaughter from the ID group, because this group only contained 4 animals (standard recommendation is that groups with <10 animals not express results in percentages, as this is a small sample size). However, in the abstract a percentage is given for the ID group at slaughter. Please re-write to exclude any percentages form the (much smaller) ID group at slaughter.
Specific comments
line 98-99: Instead of the vague "found in some lambs" be more precise here, as you were in previous lines.
line 99: I assume that whole blood was the sample used for PCR, but please state that fact.
line 119: Please write the route of administration of oxytet here.
line 156-157: The sample size is listed as 100 for these assays, but, 1 lamb died during the study. Did it die after the last sampling timepoint? Please specify when the lamb died and ensure that the sample sizes are correct.
line 166: Approximately when were the lambs sent to slaughter? How long after the last sampling timepoint?
Author Response
Dear reviewer,
Thank you very much for your comments that clearly improve the manuscript.
We answer your comments below.
General comments
- Please include the reference ranges for hematocrit, reticulocytes, and monocytes in the text of the manuscript, not just the figure legends. These have been added.
- Please be specific in the route of administration of "injectable" drugs. In your discussion you finally specify that the route is IM, but, in much of the manuscript, the route is simply listed as "injectable" which is too general for your readers. This information has been added to M&M.
- Throughout much of the manuscript, you have been careful to not write a percentage for data obtained at slaughter from the ID group, because this group only contained 4 animals (standard recommendation is that groups with <10 animals not express results in percentages, as this is a small sample size). However, in the abstract a percentage is given for the ID group at slaughter. Please re-write to exclude any percentages form the (much smaller) ID group at slaughter. This has been modified.
Specific comments
line 98-99: Instead of the vague "found in some lambs" be more precise here, as you were in previous lines. Data added.
line 99: I assume that whole blood was the sample used for PCR, but please state that fact. This information was already in the manuscript in line 136.
line 119: Please write the route of administration of oxytet here. This was added.
line 156-157: The sample size is listed as 100 for these assays, but, 1 lamb died during the study. Did it die after the last sampling timepoint? Please specify when the lamb died and ensure that the sample sizes are correct. This has been clarified.
line 166: Approximately when were the lambs sent to slaughter? How long after the last sampling timepoint? This information was given in the first paragraph of M&M. Lambs were sent to the abattoir with 2.5-3 months, so the last sampling was done almost immediately before slaughter.
Thank you very much and kind regards.
Reviewer 2 Report
Comparative study of the use of doxycycline and oxytetracycline to treat anaplasmosis in fattening lambs
Review
It is an interesting paper and it deserves publication. However, the part M and M is redundant written and details must be provided.
Introduction
History of Anaplasmosis in Spain
Prevalence of ticks in Spain and Europe (there are available reviews)
Update the literature.
The whole paper can be extended and make it complete paper suitable to be published in the relevant journal
Material and methods
Map, geography,
Bacterial-Parasitological examinations
Biomolecular exams, pl provide details.
LM images from all of the detected pathogens and blood cells.
Results
Provide info of the biomolecular analysis
Provide images of the examined cells and pathogens
330-333
It is well known that the main biological vectors of anaplasmosis are ticks belonging to the genera Rhipicephalus (i.e., Rhipicephalus bursa and Rhipicephalus sanguineus sensu lato), Dermacentor (i.e. Dermacentor silvarum, Dermacentor marginatus and Dermacentor andersoni), Ixodes and Amblyomma [3,4,5, 20, 21]
The species names pl in cursive
References
Please update and complete the international literatures research
Pl compare same studies in Japan regarding doxycycline and oxytetracycline
Pl edit all the refs and present based on the Journals guidelines.
Author Response
Dear reviewer,
Thank you very much for your coments that have been answered below.
Introduction
History of Anaplasmosis in Spain. Some information has been added. However, unfortunately, there are no epidemiological data on the importance of anaplasmosis in Spain. The information offered is based on our personal experience in the clinical service we offer at the Vet Faculty of Zaragoza, where we receive between 400 and 500 cases referred each year. Since it was diagnosed for the first time in 2014, the number of cases received and the consultations that have been made to us by telephone and electronically on the subject have been increasing year after year.
Prevalence of ticks in Spain and Europe (there are available reviews). New information has been added.
Update the literature.
The whole paper can be extended and make it complete paper suitable to be published in the relevant journal
Material and methods
Map, geography, A map with the location of the farm has been added.
Bacterial-Parasitological examinations. Although some smears were checked to locate the bacteria within the erythrocytes, this technique was not used with diagnostic proposes as it became obsolete after the use of molecular diagnostic methods.
Biomolecular exams, pl provide details. We think that these details are clearly exposed in section 2.3 of M&M section.
LM images from all of the detected pathogens and blood cells. Although we have some pictures from these animals, we think that is not relevant for the paper. As it was said before, the main diagnostic method used was PCR.
Results
Provide info of the biomolecular analysis. It has been provided.
Provide images of the examined cells and pathogens. As it was explained before, we do not find it relevant for the paper.
330-333
It is well known that the main biological vectors of anaplasmosis are ticks belonging to the genera Rhipicephalus (i.e., Rhipicephalus bursa and Rhipicephalus sanguineus sensu lato), Dermacentor (i.e. Dermacentor silvarum, Dermacentor marginatus and Dermacentor andersoni), Ixodes and Amblyomma [3,4,5, 20, 21]
The species names pl in cursive. It has been modified.
References
Please update and complete the international literatures research
Pl compare same studies in Japan regarding doxycycline and oxytetracycline
Pl edit all the refs and present based on the Journals guidelines.
It has been updated.
Thank you very much and kind regards.